# ABO-Incompatible Liver Transplantation under the Desensitization Protocol with Rituximab: Effect on Biliary Microbiota and Metabolites

**DOI:** 10.3390/jcm12010141

**Published:** 2022-12-24

**Authors:** Min Xiao, Zhenmiao Wan, Xin Lin, Di Wang, Zhitao Chen, Yangjun Gu, Songming Ding, Shusen Zheng, Qiyong Li

**Affiliations:** 1Department of Surgery, Shulan (Hangzhou) Hospital Affiliated to Zhejiang Shuren University Shulan International Medical College, Hangzhou 310004, China; 2Jinan Microecological Biomedicine Shandong Laboratory, Jinan 250021, China; 3Division of Hepatobiliary and Pancreatic Surgery, Zhejiang Chinese Medical University, Hangzhou 310053, China

**Keywords:** bile microbiota, 16S rRNA amplicon sequencing, ABO incompatibility, liver transplantation

## Abstract

Background: ABO-incompatible liver transplantation (ABOi LT) under the desensitization protocol with rituximab had excellent survival outcomes comparable to those of ABO-compatible liver transplantation (ABOc LT). In this work, we explored the effect of ABOi LT on recipients from the perspective of biliary microbiota and metabonomics. Methods: Liver transplant (LT) recipients treated at our center were enrolled in the study. In total, 6 ABOi LT recipients and 12 ABOc LT recipients were enrolled, and we collected their bile five times (during LT and at 2 days, 1 week, 2 weeks and 1 month after LT). The collected samples were used for 16S ribosomal RNA sequencing and liquid chromatography mass spectrometry analysis. Results: We obtained 90 bile samples. Whether in group ABOi LT or ABOc LT, the most common phyla in all of the samples were *Firmicutes*, *Proteobacteria*, *Bacteroidetes* and *Actinobacteria*. The most common genera were *Lactobacillus*, *Weissella*, *Klebsiella*, *Pantoea* and *Lactococcus*. There was no significant difference in the diversity between the two groups at 1 week, 2 weeks and 1 month after LT. However, the biggest disparities between the ABOi LT recipients and ABOc LT recipients were observed 2 days after LT, including increased biodiversity with a higher ACE, Chao1, OBS and Shannon index (*p* < 0.05), and more Staphylococcus in ABOi LT and binary–Jaccard dissimilarity, which indicated varying β-diversity (*p* = 0.046). These differences were not observed at 1 week, 2 weeks and 1 month after LT. The principal coordinate analysis (PCoA) revealed that the composition of the bile microbiota did not change significantly within 1 month after LT by longitudinal comparison. In an analysis of the bile components, the metabolites were not significantly different every time. However, four enrichment KEGG pathways were observed among the groups. Conclusion: These findings suggest that ABOi LT under the desensitization protocol with rituximab did not significantly affect the biliary microbiota and metabolites of recipients.

## 1. Introduction

Liver transplantation (LT) is the best treatment option for patients with end-stage liver disease, such as liver cirrhosis and liver tumor, as well as acute liver failure. Survival rates after LT today are excellent. However, the shortage of donor organs restricts the development of LT and results in a significant risk of waitlist mortality [1]. Patients in urgent need of liver transplantation are likely to die due to the lack of suitable donors. To widen donor access for an acute situation, ABO-incompatible LT (ABOi LT) can play a significant role in such patients. Nevertheless, in ABOi LT, antibodies to the donor blood group’s antigen induce severe rejection with a high rate of complications, including hepatic necrosis, extensive intrahepatic biliary tract destruction and diffuse intravascular coagulation disorder within the graft, which lead to poor graft and patient survival [2,3].

Nowadays, with the advent of strategies to overcome the ABO blood group barrier, the 3 year survival rate in patients who have had an ABOi LT is as high as 73.3%, which is comparable with that in ABO-compatible LT (ABOc LT) recipients [4]. Therefore, ABOi LT has been a routine procedure that is an effective and safe way to extend the lifespan [5]. With the introduction of rituximab (RTX), the incidence of antibody-mediated rejection of ABOi LT has been markedly reduced [6]. Another study also showed that the ABOi LT with rituximab prophylaxis had excellent survival outcomes, comparable to those of ABOc LT, except that the incidence of biliary stricture was significantly higher [7]. 

Human microecology is a special ecosystem that forms during the long-term interaction between microorganisms and hosts and plays an important role in human health and disease. At present, research on the microecology of the digestive system mainly focuses on the gut, but there may be unique microecological systems in the biliary system. However, there is little research on both the healthy and pathological biliary systems. Researchers have explored the relationship between biliary microbiota and the occurrence, development, diagnosis and treatment of hepatobiliary diseases, such as gallbladder stones [8], common bile duct stones [9], sphincter of Oddi damage and the recurrence of common bile duct stones [10], primary sclerosing cholangitis [11] and biliary tract tumors [12].

Two other studies have investigated biliary microbiota in LT patients with biliary complications [8,13]. Significant changes occurred in the biliary microbiota in patients with biliary complications after LT. At present, there has been no comparative study on the biliary microbiota of ABOi and ABOc LT. As multiple bacteria coexist in the biliary tract, culture-dependent methods are insensitive to bacterial identification and are inadequate for studying the entire microbial community [14,15]. High-throughput sequencing technology reveals a microbial-rich composition and promotes the discovery of new biliary bacteria, which can enrich our knowledge of the impact on prognoses in patients with ABOi LT. In this study, high-throughput 16S ribosomal RNA (16S rRNA) gene sequencing and liquid chromatography mass spectrometry (LC–MS) were used to compare the bacterial communities and metabolomics in the bile of patients after ABOi LT versus ABOc LT. 

## 2. Materials and Methods

### 2.1. Study Population

The patients enrolled in the study were LT recipients treated at our center from September 2020 to February 2021. The patients were included in the ABOi LT group if they met the following criteria: (1) presentation at our center and receipt of ABOi LT, (2) aged 18–65 years and (3) T-tube indwelled to drain bile. Exclusion was based on the following criteria: (1) history of gallbladder and biliary tract surgery before LT or intraoperative biliary tract injury, (2) not-first LT and (3) failure of LT or death within 1 month after surgery. The patients who underwent ABOc LT were included in a control group to compare outcomes. Clinical metadata, including age, gender, body mass index (BMI), previous abdominal surgery, blood test results and transplantation data, were collected. In the end, 6 recipients were in the ABOi LT group, and 12 recipients were in the ABOc LT group. 

The desensitization protocol for ABO incompatibility was a single dose of rituximab (300–375 mg/m^2^ per body surface area). Rituximab was infused intravenously 2–3 h before and during LT. Post-transplant immunosuppression comprised tacrolimus, basiliximab, corticosteroid and mycophenolate mofetil (MMF). This study was approved by the Ethics Committee of Shulan (Hangzhou) Hospital. Written informed consent was obtained from the participants. Organs from executed prisoners were not used in this study. In addition, prophylactic antibiotics were used to prevent infection 24 h before and after surgery.

### 2.2. Sample Collection and Storage

Bile samples were collected following a strict protocol to ensure aseptic conditions and to avoid possible microbial contamination. A T-tube was routinely inserted during the liver transplantations, and the procedure was as follows: the donor common bile duct and the recipient common bile duct were anastomosed end to end, the posterior wall was sutured continuously, the forearm was sutured intermittently and a 12# rubber T-tube was inserted. We did not perform bilioenteric anastomosis. 

Bile was collected five times from every patient at different times: During LT and after abdominal exploration but before starting the hepatectomy, at least 5 mL bile was aspirated from the gallbladder with a sterile 20 mL syringe. After LT, at least 5 mL bile was collected aseptically at 2 days, 1 week, 2 weeks and 1 month from a T-tube. The bile samples were stored at −80 ℃ until used for the DNA extraction. 

### 2.3. Definition of the Sample Groups

The collected samples were classified into the following ten groups according to whether the blood type between the donor and recipient was compatible and the time of the sample collection: (1) group ABOi-0—samples collected during ABOi LT, n = 6; (2) group ABOi-1—samples collected 2 days after ABOi LT, n = 6; (3) group ABOi-2—samples collected 1 week after ABOi LT, n = 6; (4) group ABOi-3—samples collected 2 weeks after ABOi LT, n = 6; (5) group ABOi-4—samples collected 1 month after ABOi LT, n = 6; (6) group ABOc-0—samples collected during ABOc LT, n = 12; (7) group ABOc-1—samples collected 2 days after ABOc LT, n = 12; (8) group ABOc-2—samples collected 1 week after ABOc LT, n = 12; (9) group ABOc-3—samples collected 2 weeks after ABOc LT, n = 12; (10) group ABOc-4—samples collected 1 month after ABOc LT, n = 12.

### 2.4. DNA Extraction, Illumina MiSeq Sequencing and Bioinformatic Analysis

The bacterial DNA was isolated from the bile samples using a MagPure Soil DNA LQ Kit (Magen, Guangdong, China). The DNA concentration and integrity were measured using a NanoDrop 2000 spectrophotometer (Thermo Fisher Scientific, Waltham, MA, USA) and agarose gel electrophoresis, respectively. The PCR amplification of the V3-V4 hypervariable regions of the bacterial 16S rRNA gene was carried out in a 25 μL reaction using universal primer pairs (343F: 5’-TACGGRAGGCAGCAG-3’; 798R: 5’AGGGTATCTAATCCT-3’). The reverse primer contained a sample barcode and both primers were connected with an Illumina sequencing adapter. 

The amplicon quality was visualized using gel electrophoresis. The PCR products were purified with Agencourt AMPure XP beads (Beckman Coulter Co., Brea, CA, USA) and quantified using a Qubit dsDNA assay kit. Subsequently, the concentrations were adjusted for sequencing. The sequencing was performed on an Illumina NovaSeq6000, with two paired-end read cycles of 250 bases each (Illumina Inc., San Diego, CA, USA; OE Biotech Company, Shanghai, China).

The raw sequencing data were in the FASTQ format. The paired-end reads were then preprocessed using Cutadapt software to detect and cut off the adapter. After trimming, the paired-end reads were filtered for low-quality sequences; denoised, merged and detected; and the chimera reads cut off using DADA2 [16] with the default parameters of QIIME2. Finally, the software output the representative reads and the amplicon sequence variant (ASV) abundance table. The representative read of each ASV was selected using the QIIME 2 package [17]. All representative reads were annotated and blasted against the Silva database Version 138 (or Unite; 16s/18s/ITS rDNA) using a q2feature classifier with the default parameters. 

All statistical analyses were conducted using the R programming language. Species’ α-diversity was evaluated using the ACE, Chao1, OBS, Shannon and Simpson indices. ACE, Chao1 and OBS are indices used to access species richness: the greater the value in these indices, the higher the richness of the microbial community. Shannon and Simpson are common indices for measures of species diversity and evenness. The greater the indices, the higher the diversity and the more even the distribution. The differences in bile microbiota among the habitats could be evaluated by β-diversity analysis, which was evaluated with principal coordinate analysis (PcoA). The calculations of PcoA were performed by binary–Jaccard distance, Bray–Curtis distance and (un)weighted UniFrac distance. The LDA effect size (Lefse) analysis was performed to identify species in which the relative abundance significantly diverged among populations. We identified differential distributions of species using the default cut-offs (LDA score < 2.0; *p*-value < 0.05). PICRUSt was used to the predict bacterial function through the Kyoto Encyclopedia of Genes and Genomes (KEGG) database. A Wilcoxon rank-sum test was performed to evaluate the disparities among the populations in alpha diversity, principal coordinates and community difference analysis. A Kruskal–Wallis test was used to assess the changes in biliary tract microecology within 1 month after LT. A *p* < 0.05 was required for the results to be considered statistically significant. 

### 2.5. Analysis of Targeted Metabolomics

High-performance LC–MS was performed to profile the composition of the bile samples. All of the chemicals and solvents were of analytical or HPLC grade. Water, methanol, acetonitrile and formic acid were purchased from Thermo Fisher Scientific (Waltham, MA, USA). L-2-chlorophenylalanine was obtained from Shanghai Heng Chuang Bio-technology Co., Ltd. (Shanghai, China). Chloroform was purchased from Titan Chemical Reagent Co., Ltd. (Shanghai, China). The bile samples stored at −80 °C were thawed at room temperature. The bile was added to a 1.5 mL Eppendorf tube with L-2-chlorophenylalanine (0.3 mg/mL) dissolved in methanol as an internal standard, and the tube was vortexed for 10 s. Subsequently, an ice-cold mixture of methanol and acetonitrile was added, and the mixtures were vortexed for 1 min. Whole samples were extracted by ultrasonic for 10 min in an ice-water bath and stored at −20 °C for 30 min. The extract was centrifuged (4 °C at 13,000 rpm) for 10 min. The supernatants from each tube were collected using crystal syringes, filtered through 0.22 μm microfilters and transferred to LC vials. The vials were stored at −80 °C until the LC–MS analysis.

The ACQUITY UPLC I-Class system (Waters Corporation, Milford, MA, USA) coupled with a VION IMS QTOF Mass Spectrometer (Waters Corporation, Milford, MA, USA) was used to analyze the metabolic profiling in both the ESI-positive and ESI-negative ion modes. An ACQUITY UPLC BEH C18 column (1.7 μm, 2.1 × 100 mm) was employed in both the positive and negative modes. Water and acetonitrile/methanol, 2/3 (*v*/*v*), both containing 0.1% formic acid, were used as mobile phases A and B, respectively. The linear gradients were as follows: 0 min, 1% B; 1 min, 30% B; 2.5 min, 60% B; 6.5 min, 90% B; 8.5 min, 100% B; 10.7 min, 100% B; 10.8 min, 1% B; and 13 min, 1%B. The flow rate was 0.4 mL/min, and the column temperature was 45 °C. All samples were kept at 4 °C during the analysis. The injection volume was 1 μL. The data acquisition was performed in the full-scan mode (*m*/*z* ranges from 50 to 1000) combined with the MSE mode, including 2 independent scans with different collision energies (Ces) that were alternatively acquired during the run. The parameters of the mass spectrometry were as follows: a low-energy scan (CE 4 eV) and a high-energy scan (CE ramp 20–45 eV) to fragment the ions. Argon (99.999%) was used as a collision-induced dissociation gas, with a scan time: 0.2 s; interscan delay: 0.02 s; capillary voltage: 2.5 kV; cone voltage: 40 V; source temperature: 115 °C; desolvation gas temperature: 450 °C; and desolvation gas flow: 900 L/h. The QCs were injected at regular intervals (every bile sample) throughout the analytical run to provide a set of data from which the repeatability could be assessed.

The matrix was imported into R for principal component analysis (PCA) to observe the overall distribution among the samples and the stability of the whole analysis process. Orthogonal partial least squares discriminant analysis (OPLS–DA) was utilized to distinguish metabolites between the groups. To prevent overfitting, 7-fold cross-validation and 200 response permutation testing (RPT) were used to evaluate the quality of the model. The variable importance of projection (VIP) values obtained from the OPLS–DA model were used to rank the overall contribution of each variable to group discrimination. A two-tailed Student’s *t*-test was further used to verify whether the metabolites of difference between the groups were significant. The differential metabolites were selected with VIP values greater than 1.0 and *p*-values less than 0.05. Finally, the differential metabolites were annotated through the metabolic pathways in the KEGG database (https://www.kegg.jp/kegg/pathway.html (accessed on 15 December 2021)) to obtain the pathways involved in the differential metabolites. 

### 2.6. Data Analysis

Quantitative demographic and clinical data with two normal distributions were expressed as mean deviations (minimum–maximum) and analyzed by Student’s *t*-test. The Wilcoxon rank-sum test was used to analyze non-normally distributed quantitative sequencing data. SPSS software (Version 25.0, IBM SPSS Inc., Chicago, USA) was used for analyses. Two-tailed *p* < 0.05 was considered to indicate a statistically significant difference.

## 3. Results

### 3.1. Patient Characteristics and 16S rRNA Sequencing

From September 2020 to February 2021, 6 ABOi LT recipients and 12 ABOc LT recipients were included in the study. The primary diseases included hepatocellular carcinoma, hepatitis B cirrhosis, liver failure and colorectal liver metastasis. We collected blood test results 1 day before liver transplantation in all patients. The characteristics of these patients are shown in Table 1. The clinical variables were balanced between both groups, except white blood cell count, blood platelet, C-reactive protein and cold ischemia time. The elevation of the white blood cell count and the level of C-reactive protein suggests a more active inflammatory response and worse liver function in the patients with ABOi LT. Among the 18 LT recipients, only one patient in the ABOi LT group developed biliary fistula at 1 month, and the rest had no adverse clinical events, such as rejection, biliary complications and graft loss within 1 month.

A total of 90 samples were submitted for 16S rRNA sequencing. The data volume of the raw reads after sequencing was between 73,381 and 81,982, and the volume of clean tags after the quality control was between 64,176 and 73,154. The data volume of the valid tags obtained by the clean tags after removing the chimera was between 59,084 and 69,104, and the number of ASVs in each sample was distributed between 208 and 1095. A total of 25,109 ASVs were identified and assigned taxonomic ranks in these samples. According to the ASV division and classification status, the microbial composition of the samples at each classification level could be obtained, and a total of 28 phyla and 701 genera were identified in this work. 

### 3.2. Comparison between Biliary Microbiota of ABOi LT and ABOc LT at Diverse Times 

#### 3.2.1. Two Days after LT

Only 654 ASVs were shared on ABOi-1 and ABOc-1 (Figure 1A). According to the average relative abundance, in the ABOi-1 group, the most common phyla were *Firmicutes* (55.62%), *Proteobacteria* (32.38%), *Bacteroidetes* (7.28%) and *Actinobacteria* (1.35%), and the most common genera were *Lactobacillus* (23.77%), *Weissella* (18.32%), *Klebsiella* (6.76%) and *Pantoea* (6.04%). In the ABOc-1 group, the most common phyla were *Firmicutes* (50.13%), *Proteobacteria* (39.08%), *Bacteroidetes* (6.10%) and *Actinobacteria* (1.28%). The most common genera were *Lactobacillus* (21.14%), *Klebsiella* (17.85%), *Weissella* (16.85%) and *Pantoea* (5.50%; Figure 1B,C). After the α-diversity analysis of groups ABOi-1 and ABOc-1, we found that the ACE, Chao1 and OBS indices were higher in the ABOi-1 group (*p* < 0.05). The Shannon index in the ABOi-1 group was significantly higher than that in ABOc-1 (*p* < 0.05), but there were no statistically significant differences in the Simpson index between the two groups (*p* > 0.05; Figure 1D). The difference in the bile microbiota between the ABOi-1 and ABOc-1 groups could be evaluated by the β-diversity. A PcoA based on binary–Jaccard dissimilarity indicated that the first and second principal components (PC1 and PC2) contributed 6.71% and 6.39% of the total variance between groups ABOi-1 and ABOc-1 (Figure 1E, *p* = 0.046). The other three distances (Figure 1E, *p* = 0.359, *p* = 0.724 and *p* = 0.453) revealed that the two groups could not be completely discriminated against. Additionally, the significantly different bacteria between the ABOi-1 group and ABOc-1 group at the genus level are shown in Figure 1F. The proportion of *Burkholderia–Caballeronia–Paraburkholderia*, *Bacteroides*, *Prevotella*, *Rikenellaceae RC9 gut group*, *Staphylococcus*, *Bacteriovorax*, *Moheibacter* and *Shinella* increased significantly, and the abundance of *67–14* and *Acidovorax* decreased markedly in the ABOi-1 group (LDA > 2). The results of the Lefse analysis showed the class, order, family, genus or species with significantly distinct abundances between the ABOi-1 and ABOc-1 groups (Figure 1G).

There were disparities between the two groups at Level 2 of the KEGG pathway. We identified 11 significantly enriched pathways in the bile samples in the ABOi-1 group versus the ABOc-1 group. The top six significantly enriched pathways in the ABOi-1 group were related to neurodegenerative diseases, cardiovascular diseases, cell motility, endocrine and metabolic diseases, drug resistance: antineoplastic, and cell growth and death (Figure 1H) at Level 3 of the KEGG pathway. We identified 69 significantly enriched pathways in the bile samples of the ABOi-1 group. The top six significantly enriched pathways in the ABOi-1 group related to the glyoxylate and dicarboxylate metabolism, fatty acid metabolism, butanoate metabolism, oxidative phosphorylation, quorum sensing and the two-component system (Figure 1I). 

In the metabolite analysis, the results of the PCA (Figure 1J) and OPLS–DA (Figure 1K) indicated that there was no difference in metabolites between the ABOi-1 and ABOc-1 groups. We did not screen for varying metabolites between the two groups (Figure 1L). Four enriched KEGG pathways were obtained between the groups, including primary bile acid biosynthesis, bile secretion, cholesterol metabolism and taurine and hypotaurine metabolism (Figure 1M). 

#### 3.2.2. One Week after LT

In total, 652 ASVs were shared on ABOi-2 and ABOc-2 (Figure 2A). According to the average relative abundance, in the ABOi-2 group, the most common phyla were *Firmicutes* (58.82%), *Proteobacteria* (29.42%), *Bacteroidetes* (6.94%) and *Actinobacteria* (1.21%), and the most common genera were *Lactobacillus* (22.32%), *Weissella* (18.01%), *Klebsiella* (6.48%) and *Pantoea* (5.96%). In the ABOc-2 group, the most common phyla were *Firmicutes* (56.51%), *Proteobacteria* (31.12%), *Bacteroidetes* (7.42%) and *Actinobacteria* (1.25%). The most common genera were *Lactobacillus* (24.04%), *Weissella* (19.15%)*, Klebsiella* (6.57%) and *Pantoea* (6.37%; Figure 2B,C). After the α-diversity analysis of the ABOi-2 and ABOc-2 groups, we found that the differences in the α-diversity between ABOi-2 and ABOc-2 were not significant (Figure 2D). Likewise, there was no marked disparity in the β-diversity analysis (Figure 2E, *p* = 0.09, *p* = 0.326, *p* = 0.772 and *p* = 0.258). Additionally, the top 10 significantly different bacteria between groups ABOi-2 and ABOc-2 at the genus levels are displayed in Figure 2F. The proportion of *Kineococcus*, *Anaerotruncus* and *Woeseia* increased significantly, and the abundance of the *Lachnospiraceae NK4A136 group*, *Roseburia*, *Pseudoxanthomonas*, *Limibaculum*, *Dokdonella*, *Limnochordaceae* and *Macrococcus* decreased markedly in the ABOi-2 group. The results of the LEfse analysis revealed the class, order, family, genus or species with a markedly different abundance between groups ABOi-2 and ABOc-2 (Figure 2G). 

There was no notable variation between the two groups at Level 2 of the KEGG pathway. At Level 3 of the KEGG pathway, we identified one noticeably depleted pathway in the bile samples of the ABOi-2 group, which was indole alkaloid biosynthesis (Figure 2H). 

In the metabolite analysis, the results of the PCA (Figure 2I) and OPLS-DA (Figure 2J) indicate that there was no distinction in the metabolites between the ABOi-2 and ABOc-2 groups. We did not screen for varying metabolites between the two groups (Figure 2K). Four enriched KEGG pathways were obtained between groups, including primary bile acid biosynthesis, bile secretion, cholesterol metabolism, taurine and hypotaurine metabolism (Figure 2L). 

#### 3.2.3. Two Weeks after LT

In total, 712 ASVs were shared on ABOi-3 and ABOc-3 (Figure 3A). According to the average relative abundance, in the ABOi-3 group, the most common phyla were *Firmicutes* (54.81%), *Proteobacteria* (29.82%), *Bacteroidetes* (9.51%) and *Actinobacteria* (1.34%), and the most common genera were *Lactobacillus* (22.01%), *Weissella* (17.15%), *Lactococcus* (5.76%) and *Klebsiella* (5.48%). In the ABOc-3 group, the most common phyla were *Firmicutes* (55.87%), *Proteobacteria* (31.05%), *Bacteroidetes* (7.86%) and *Actinobacteria* (1.43%). The most common genera were *Lactobacillus* (23.44%), *Weissella* (18.86%), *Klebsiella* (6.32%) and *Pantoea* (6.27%; Figure 3B,C). After the α-diversity analysis of the groups ABOi-3 and ABOc-3, we found that the variations in the α-diversity between the two groups were not significant (Figure 3D). Similarly, there was no significant difference in the β-diversity analysis (*p* = 0.174, *p* = 0.08, *p* = 0.239 and *p* = 0.234) between the two groups (Figure 3E). Additionally, the top 10 significantly different bacteria between the two groups at the genus level are shown in Figure 3F. The proportion of *Collinsella* and *Ralstonia* increased significantly, and the abundance of *Pantoea*, *Salinimicrobium*, *Allobaculum*, *Lonsdalea*, *Shinella*, *Thiohalomonas*, *Gluconobacter* and *Woeseia* decreased markedly in the ABOi-3 group. The results of the LEfse analysis revealed the class, order, family, genus or species with significantly distinct abundance between the ABOi-3 and ABOc-3 groups (Figure 3G). 

There was no significant variation between the two groups at Level 2 of the KEGG pathway. At Level 3 of the KEGG pathway, we identified two significantly depleted pathways, carotenoid biosynthesis and phosphonate and phosphonate metabolism, in the bile samples of the ABOi-3 group (Figure 3H).

In the metabolite analysis, the results of the PCA (Figure 3I) and OPLS–DA (Figure 3J) indicated that there was no difference in the metabolites between the ABOi-3 and ABOc-3 groups. We did not screen for divergent metabolites between the two groups (Figure 3K). Four enriched KEGG pathways were obtained between the groups, including primary bile acid biosynthesis, bile secretion, cholesterol metabolism, taurine and hypotaurine metabolism (Figure 3L). 

#### 3.2.4. One Month after LT

In total, 651 ASVs were shared on ABOi-4 and ABOc-4 (Figure 4A). According to the average relative abundance, in the ABOi-4 group, the most common phyla were *Firmicutes* (56.22%), *Proteobacteria* (31.35%), *Bacteroidetes* (7.34%) and *Actinobacteria* (2.11%), and the most common genera were *Lactobacillus* (23.80%), *Weissella* (18.40%), *Lactococcus* (6.51%) and *Klebsiella* (6.31%). In the ABOc-4 group, the most common phyla were *Firmicutes* (49.58%), *Proteobacteria* (39.39%), *Bacteroidetes* (6.57%) and *Actinobacteria* (1.81%). The most common genera were *Lactobacillus* (20.73%), *Weissella* (16.10%), *Klebsiella* (14.34%) and *Lactococcus* (5.52%; Figure 4B,C). After the α-diversity analysis of the ABOi-4 and ABOc-4 groups, we found that the differences in the α-diversity between the two groups were not significant (Figure 4D). Similarly, there was no significant variation in the β-diversity analysis (*p* = 0.188, *p* = 0.205, *p* = 0.909 and *p* = 0.248) between the two groups. (Figure 4E). Additionally, the top 10 significantly different bacteria between the two groups at the genus level are shown in Figure 4F. The proportion of the *(Eubacterium) siraeum group* and *Terrimonas* increased significantly, and the abundance of *Serratia*, *Pseudomonas*, *Novosphingobium*, *Staphylococcus*, *Brevundimonas*, *Odoribacter*, *Roseburia* and *Nakamurella* decreased markedly in the ABOi-4 group. The results of the LEfse analysis demonstrated the class, order, family, genus or species with a significantly divergent abundance between the ABOi-4 and ABOc-4 groups (Figure 4G). 

There was no significant difference between the two groups at Level 2 of the KEGG pathway. At Level 3 of the KEGG pathway, we identified eight significantly different pathways in the bile samples of the ABOi-4 and ABOc-4 groups (Figure 4H).

In the metabolite analysis, the results of the PCA (Figure 4I) and OPLS–DA (Figure 4J) indicated that there was no difference in the metabolites between the ABOi-4 and ABOc-4 groups. We did not screen for different metabolites between the two groups (Figure 4K). Four enriched KEGG pathways were obtained between the groups, including primary bile acid biosynthesis, bile secretion, cholesterol metabolism, taurine and hypotaurine metabolism (Figure 4L). 

### 3.3. Changes in the Biliary Microbiota within One Month after LT

We analyzed the dynamic change in the bacterial alpha diversity with the use of ACE, Chao1, Shannon and Simpson in the ABOi LT and ABOc LT groups. We found that some significantly changed for a certain period of time, while no significant change was observed in the bile collected during LT and post-LT 1 month (Figure 5A–D). Moreover, the PCoA revealed that the composition of the bile microbiota did not change significantly after LT (Figure 5E, *p* = 0.467 and *p* = 0.85). At the phylum level, we observed no microbe change in the ABOi LT group, but *Bacteroidota* changed significantly over time (Figure 5F). 

## 4. Discussion

The human microbiota is a collection of bacteria, protozoa, fungi and viruses that coexist in our bodies and are critical to the protection, metabolism and physiological functions of human health. As an important condition for the occurrence of disease, bacteria have been studied by scholars. The current research on gut microbiota has yielded meaningful results in various fields. For example, gut microbiota is related to ulcerative colitis, the improvement of autism symptoms and the relief of type 2 diabetes [18,19,20]. Liwinski [21] found ductal bile fluid owning a unique and diverse microbiota that is clearly different from oral fluid, duodenal fluid and duodenal mucosa. Therefore, the microbiota of the bile duct cannot be replaced by duodenal fluid. ABOi LT has been performed more frequently to treat acute liver disease and is a routine procedure. We explored differences in the prognostic effects of ABOc LT versus ABOi LT under the rituximab desensitization protocol by studying biliary tract microecology and metabolomics.

To the best of our knowledge, the present study is the first to use high-throughput 16S rRNA gene sequencing and LC–MS to assess variations in bile microbial composition and bile acid metabolomics between ABOi LT recipients and ABOc LT recipients. At the species level, there was considerable heterogeneity among individuals, which probably led to the result that the microecological composition of a few individual recipients differed greatly from the other in each period (Figure 1B,C, Figure 2B,C, Figure 3B,C and Figure 4B,C). In this study, we found that the most common phylum in the two groups at different times was *Firmicutes*. The other phyla with higher abundances included *Proteobacteria*, *Bacteroidetes* and *Actinobacteria*; this is similar to the control group of a previous biliary microbiota study in LT recipients [8]. At the genus level, the most dominant genus in the two groups at different times was *Lactobacillus*, followed by *Weissella*, except that in ABOc-1, *Lactobacillus* was followed by *Klebsiella*. Maybe this was because of the presence of a *Klebsiella* infection in the recipients in group ABOc-1. Zhao [22] found that RTX-induced intestinal damage may result in an imbalance of the gut microbiome and trigger an immune response. *Lactobacillus* alleviates RTX-induced inflammatory stimulation and inhibits local and systemic immune responses caused by intestinal mucosal damage. We speculate that the high abundance of *Lactobacillus* in biliary microbiota may help protect the health of the biliary tract. Our results illustrated that the main floras of the samples were similar at the phylum and genus levels between the two groups.

The precise pathophysiological mechanisms are those by which microbiota dysbiosis is often characterized by a loss of microbial diversity, reduced commensal abundance and growth of pathogenic organisms. In the ABOi-1 group, we observed an over-representation of potential pathobionts, such as *Bacteroides*, *Prevotella* and *Staphylococcus* (Figure 1F). *Staphylococcus* is a conditional pathogen, and some staphylococcal strains can produce a variety of pathogenic factors, such as exotoxins, enterotoxins, agglutination factors and biofilms [23]. The possible reason is that ABOi LT recipients are in a more acute course of the disease and have a more severe systemic inflammatory response than ABOc LT recipients. Furthermore, we observed an increased average biodiversity of the bile fluid microbiota in the ABOi-1 group (Figure 1D). From an ecological standpoint, diversity losses and gains affect ecological stability and the sustainability of ecosystem functions and services [24]. Intestinal flora studies found that a diversity index was correlated with host obesity [25] and antibiotic use [26]. In this study, there were no statistically significant differences between the two groups in BMI and preoperative antibiotic use. Therefore, it is possible that the varying usage of antibiotics led to the divergent biodiversity. Meanwhile, the binary–Jaccard dissimilarity indicated that they had different microbiota (Figure 1E, *p* = 0.046). The other three indices showed no difference. We believe that the wrong *p*-value of the binary–Jaccard may be caused by the small sample size. Overall, 2 days after LT, the two groups had similar microbiota, but the ABOi group had higher biodiversity. In the ABOi-2 group, we observed a significant decrease in *Roseburia* (Figure 2F). Studies have confirmed that *Roseburia* participates in the production of short-chain fatty acids (SCFAs), which regulate the colonic pH and reduce colonic inflammation [27,28,29,30]. This effect may also exist in the biliary tract. In the ABOi–3 group, we observed a notable decrease in *Pantoea* (Figure 3F), which was often considered a plant pathogen, but recent evidence suggests that Pantoea is often isolated from the hospital environment, and there is considerable controversy regarding its role in human disease [31]. Further studies are needed to explore this effect. In the ABOi-4 group, we observed a significant increase in Serratia (Figure 4G); some biotypes of *Serratia* can produce prodigiosin, which has immunosuppressive and antitumor properties that may be of high clinical value [32]. Thus, we believe that the potential functional and prognostic role of *Serratia* in post-LT recipients should be studied within the bile ducts in future follow-up studies. Moreover, we found an important increase in staphylococcus again. This is because one recipient in the ABOi LT group developed biliary complications 1 month after surgery, including anastomotic stenosis, anastomotic fistula and common bile duct dilation. Diversity analyses at 1 week, 2 weeks and 1 month after LT (Figure 2D,E, Figure 3D,E and Figure 4D,E) revealed no notable disparity in the biliary microbiota of ABOi LT and ABOc LT. 

After LT, we observed that there may be transient changes in the microbiota α-diversity within 1 month, but eventually, it would return to the initial microecological α-diversity (Figure 5A–D). Moreover, β-diversity does not change over time (Figure 5E). At the phylum level, the microbes in the ABOi LT group did not change over time, but *Bacteroidetes* in the ABOc LT group did (Figure 5F). One study [33] demonstrated that the tacrolimus concentrations had a significant effect on *Bacteroidetes*. It is possible that the change in the tacrolimus dose caused a change in *Bacteroidetes*. Thus, the biliary microbiota after LT is relatively stable and does not change significantly over time. The changes in the biliary flora detected over time after LT may provide clues to the occurrence and developmental mechanisms of biliary tract complications, tumor recurrence, vascular embolism and immune response. 

The occurrence of intestinal dysbiosis has been reported in post-transplant exposure to immunosuppressants, antibiotics, infection, ischemia–reperfusion injury and recurrence of the original liver disease [34,35]. This dysbiosis is also likely to occur in the biliary tract. However, due to the fact of ethical issues, there are currently a small number of studies on biliary microbiota in healthy people. This aspect of research is mainly conducted by replacing the biliary tract of healthy people with the biliary tract of LT donors and patients with pancreatic disease without hepatobiliary disease [36,37,38,39]. Therefore, it is difficult to compare biliary microbiota after transplantation with those in healthy people. In the future, the methods of obtaining and analyzing bile from healthy people must be explored and unified. Thus, the influence of LT on biliary microbiota can be described more objectively.

We analyzed the bile acid composition of the two groups collected at distinct times by LC–MS and found that there was no difference in the bile acid composition, indicating that ABOi LT had no effect on bile acid composition, and their biliary microbiota was in a similar microenvironment. Moreover, the metabolites were significantly enriched in four KEGG pathways (*p* < 0.05), including primary bile acid biosynthesis, bile secretion, cholesterol metabolism, and taurine and hypotaurine metabolism. Taurine and its derivatives can inhibit calcium precipitation in bile by decreasing the concentration of free calcium ions and increasing the solubility of unbound bilirubin [40,41]. Therefore, the decrease in the taurine content contributed to the precipitation of bilirubin calcium and the formation of calculi, which may have led to the occurrence and development of biliary stones after LT [42].

Through this study, we can gain new knowledge regarding the effects of LT on biliary microbiota. However, this experiment still has limitations. The small sample size may lead to certain deviations, and a greater sample size and multicentre research will be required to verify these findings. Due to the fact of ethical issues, it was impossible to obtain bile from patients before LT, so we used bile from patients at the beginning of LT instead. As for the method of obtaining bile through the T-tube after surgery, Andujar [43] showed that T-tube can reduce the incidence of biliary complications after LT and improve the prognosis. Therefore, routine use of rubber T-tube for choledochostomy during liver transplantation is recommended. Although the T-tube is a more direct method of obtaining bile than endoscopic retrograde cholangiopancreatography (ERCP), there is no study to compare the safety and effectiveness of the several methods of obtaining bile. Consequently, it is necessary to establish standards for the method of obtaining bile. In addition, in our study, the biliary microbiota after 1 month was not assessed, which will be conducted in a further study. All transplant recipients were given the same immunosuppressive and antibiotic regimens. The effect of antibiotics on biliary microbiota has not been clarified.

In conclusion, the present study described the characteristics of biliary microbiota post-LT and demonstrated that there was no significant difference in the bile duct microbiota between ABOi LT and ABOc LT within 1 month after LT. There are two possible reasons. First, ABOi LT itself does not cause changes in the biliary microbiota; second, ABOi LT causes changes in the biliary microbiota, and the prognosis of ABOi LT is improved under the desensitization effect of rituximab, and it may be the effect of rituximab that does not cause significant changes in biliary microbiota. This all needs to be verified by follow-up studies. The longitudinal analyses showed that the biliary microbiota after LT was relatively stable and did not change significantly over time. Moreover, there was no significant difference in the composition of the bile metabolites within 1 month after LT. Our results may provide a starting point for clinical studies on the dynamic changes in bile microbiota after LT. The findings are expected to provide valuable insights into the role of ABOi LT in rejection and other transplant-related complications.

## Figures and Tables

**Figure 1 jcm-12-00141-f001:**
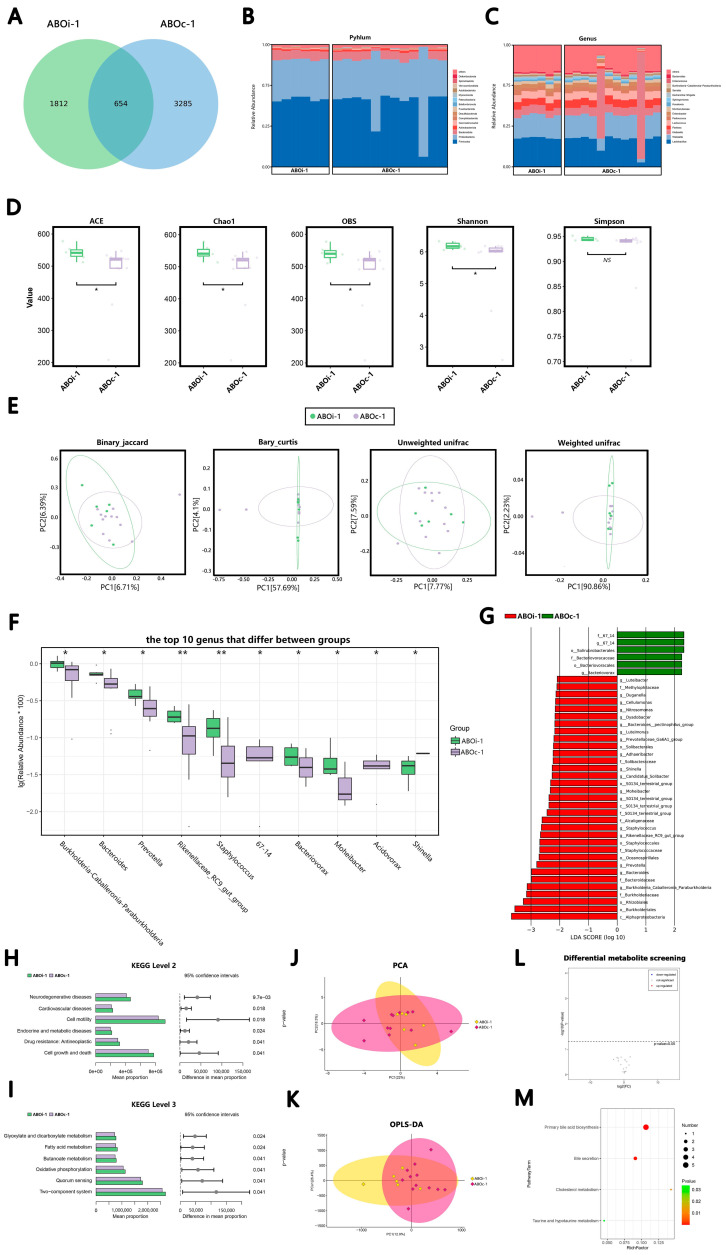
Analysis and comparison of bacterial communities and metabolomics between groups ABOi−1 and ABOc−1. (**A**) Venn diagram summarizing the number of common ASVs between the two groups. The composition and abundance distributions of the two groups at the (**B**) phylum and (**C**) genus levels. (**D**) The α-diversity (ACE, Chao1, OBS, Shannon and Simpson) between the two groups. (**E**) The β-diversity analysis was evaluated with the use of PCoA, which was calculated using the binary–Jaccard distance, Bray–Curtis distance and (un)weighted UniFrac distance. (**F**) The top 10 differential genera were detected between the two groups. For optimal visualization, a transformation of log10 (relative abundance × 100) was employed. (**G**) The histogram of the Lefse analysis revealed bacteria whose LAD score exceeded the default value of 2. Red represents the ABOi-1 group, and blue signifies the ABOc-1 group. The six significantly different KEGG pathways in (**H**) Level 2 and (**I**) Level 3 between groups ABOi-1 and ABOc-1 and (**J**) the PCA map. The distance of each coordinate point highlights the degree of aggregation and dispersion between the samples. (**K**) OPLS-DA map, where the abscissa is the predicted principal component representing the difference between the groups, and the ordinate is the orthogonal principal component, representing the difference within the group. (**L**) Volcano plot of the differential metabolites between the two groups. (**M**) KEGG pathway bubble chart of bile acid. The abscissa was the Rich factor, and the larger the value, the greater the enrichment. The ordinate signified the metabolic pathway with the highest degree of enrichment. The color of the dots stands for the *p*-value, and the redder it is, the smaller the *p*-value, indicating that the enrichment was more obvious. The size of the dot represents the number of differential genes in the pathway, and the larger the dot, the greater the number of differential genes. * *p* < 0.05 and ** *p* < 0.01.

**Figure 2 jcm-12-00141-f002:**
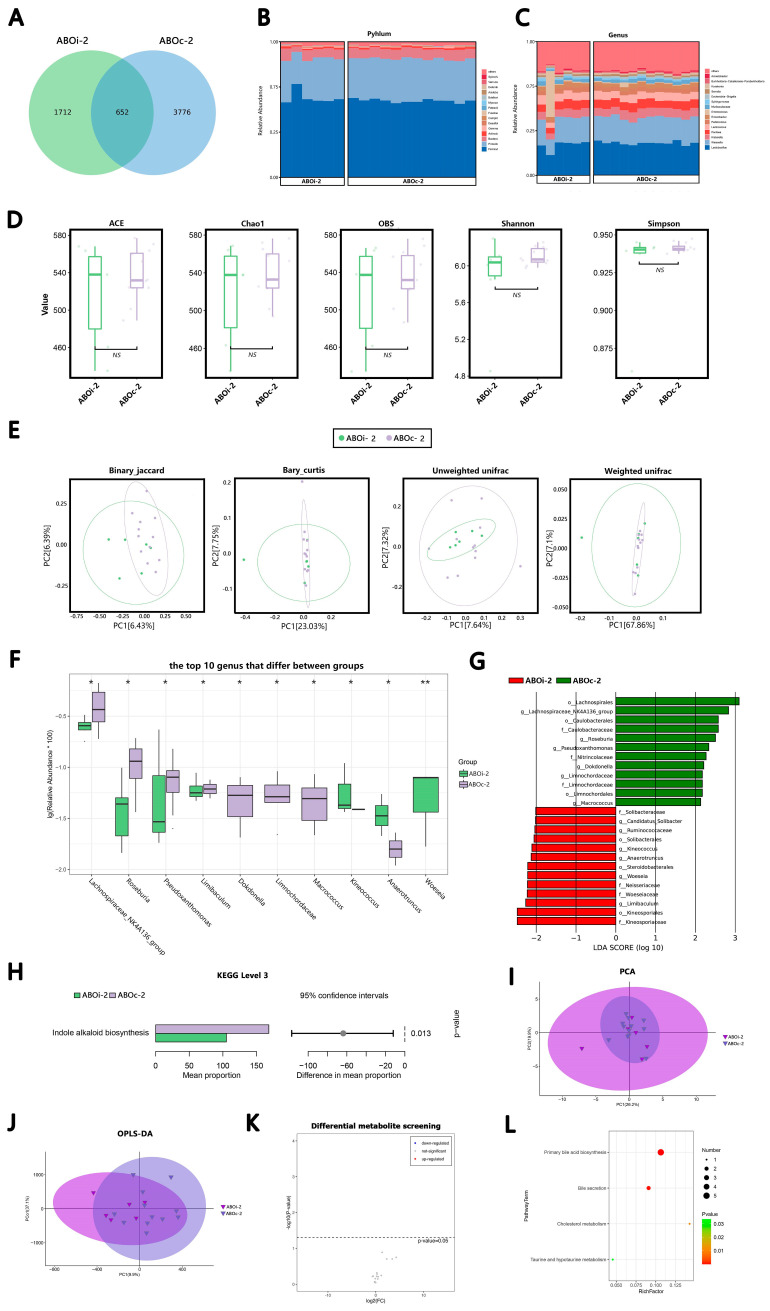
Analysis and comparison of the bacterial community and metabolomics between the groups ABOi−2 and ABOc−2. (**A**) Venn diagram summarizing the number of common ASVs between the two groups. The composition and abundance distributions of the two groups at the (**B**) phylum and (**C**) genus levels. (**D**) The α-diversity (ACE, Chao1, OBS, Shannon and Simpson) between the two groups. (**E**) The β-diversity analysis was evaluated with use of PCoA, which was calculated using the binary–Jaccard distance, Bray–Curtis distance and (un)weighted UniFrac distance. (**F**) The top 10 differential genera were detected between the two groups. For optimal visualization, a transformation of log10 (relative abundance × 100) was employed. (**G**) The histogram of LEfse analysis revealed bacteria whose LAD score exceeded the default value of 2. Red represents the ABOi-2 group, and blue represents the ABOc-2 group. (**H**) Top 6 significantly different Level 3 KEGG pathways between the two groups. (**I**) PCA map, where the distance of each coordinate point highlighted the degree of aggregation and dispersion between the samples. (**J**) OPLS-DA map, where the abscissa is the predicted principal component representing the difference between the groups, and the ordinate is the orthogonal principal component, representing the difference within the group. (**K**) Volcano plot of the differential metabolites between the two groups. (**L**) KEGG pathway bubble chart of bile acid, where the abscissa is the Rich factor, and the larger the value, the greater the enrichment. The ordinate signifies the metabolic pathway with the highest degree of enrichment. The color of the dots stands for the *p*-value, and the redder it is, the smaller the *p*-value, indicating that the enrichment was more obvious. The size of the dot represents the number of differential genes in the pathway, and the larger the dot, the greater the number of differential genes. * *p* < 0.05 and ** *p* < 0.01.

**Figure 3 jcm-12-00141-f003:**
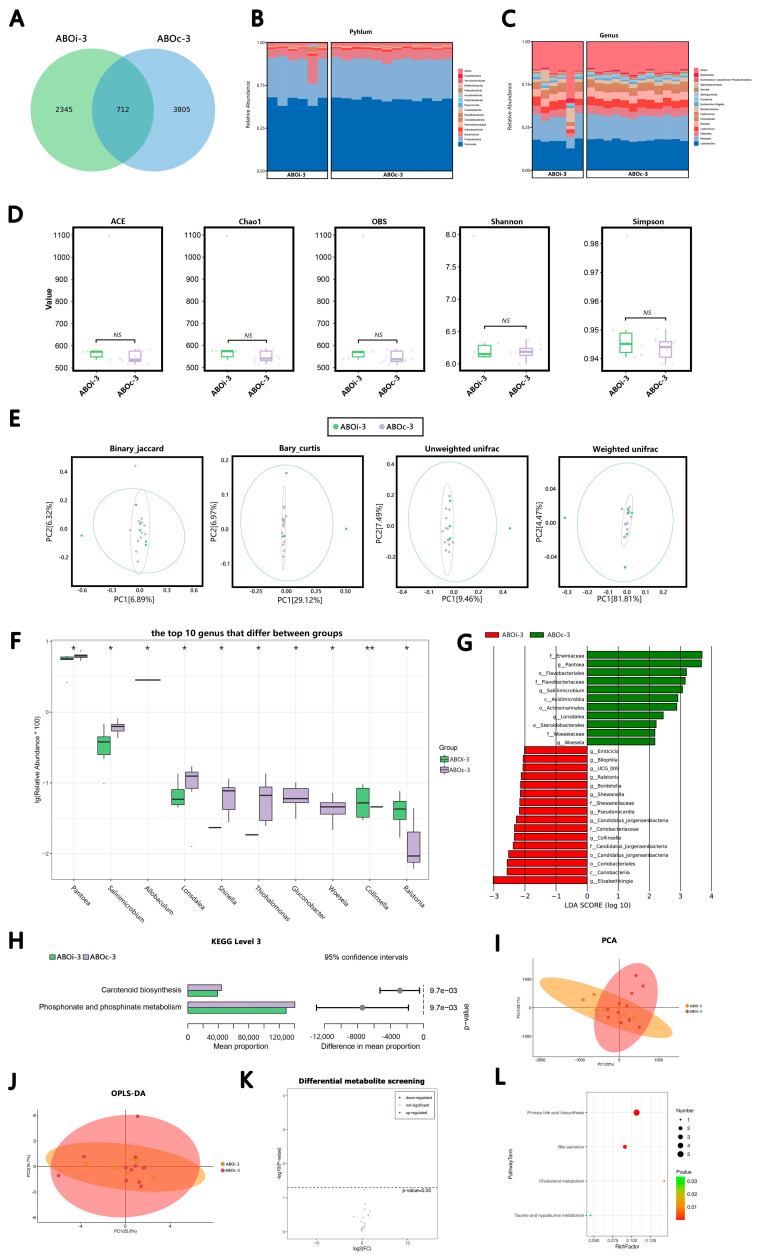
Analysis and comparison of the bacterial community and metabolomics between the the ABOi−3 and ABOc−3 groups. (**A**) Venn diagram summarizing the number of common ASVs between the two groups. The composition and abundance distributions of the two groups at the (**B**) phylum and (**C**) genus levels. (**D**) The α-diversity (ACE, Chao1, OBS, Shannon and Simpson) between two grIs. (**E**) The β-diversity analysis was evaluated with use of PCoA, which was calculated using the binary–Jaccard distance, Bray–Curtis distance and (un)weighted UniFrac distance. (**F**) The top 10 differential genera were detected between the two groups. For optimal visualization, a transformation of log10 (relative abundance × 100) was employed. (**G**) The histogram of LEfse analysis revealed bacteria whose LAD score exceeded the default value of 2. Red represents the ABOi-3 group, and blue represents the ABOc-3 group. (**H**) Top 6 significantly different Level 3 KEGG pathways between the two groups. (**I**) PCA map, where the distance of each coordinate point highlighted the degree of aggregation and dispersion between the samples. (**J**) OPLS-DA map, were the abscissa was the predicted principal component representing the difference between the groups, and the ordinate was the orthogonal principal component, representing the difference within the group. (**K**) Volcano plot of the differential metabolites between the two groups. (**L**) KEGG pathway bubble chart of bile acid, where the abscissa is the Rich factor, and the larger the value, the greater the enrichment. The ordinate signifies the metabolic pathway with the highest degree of enrichment. The color of the dots stands for the *p*-value, and the redder it is, the smaller the *p*-value, indicating that the enrichment was more obvious. The size of the dot represents the number of differential genes in the pathway, and the larger the dot, the greater the number of differential genes. * *p* < 0.05 and ** *p* < 0.01.

**Figure 4 jcm-12-00141-f004:**
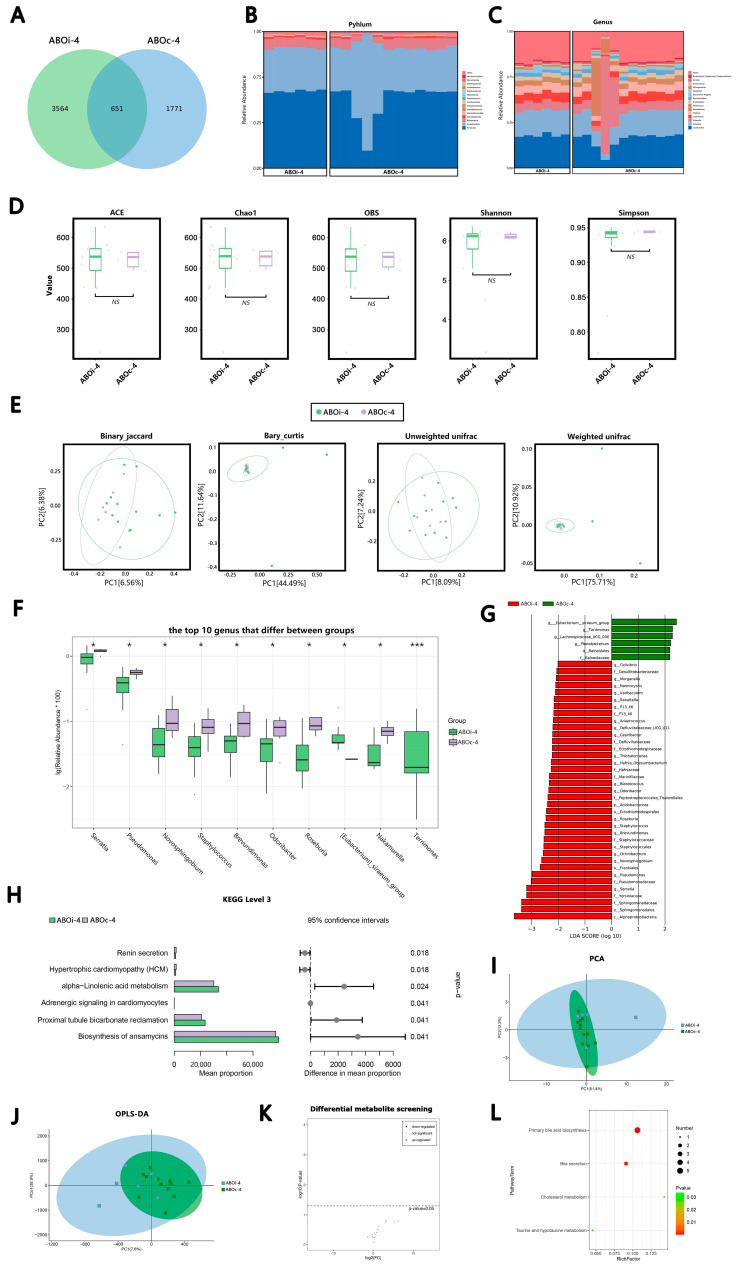
Analysis and comparison of the bacterial community and metabolomics between the ABOi−4 and ABOc-4 groups. (**A**) Venn diagram summarizing the number of common ASVs between the two groups. The composition and abundance distributions of the two groups at the (**B**) phylum and (**C**) genus levels. (**D**) The α-diversity (ACE, Chao1, OBS, Shannon and Simpson) between the two groups. (**E**) The β-diversity analysis was evaluated with use of PCoA, which was calculated using the binary–Jaccard distance, Bray–Curtis distance and (un)weighted UniFrac distance. (**F**) The top 10 differential genera were detected between the two groups. For optimal visualization, a transformation of log10 (relative abundance × 100) was employed. (**G**) The histogram of the LEfse analysis revealed bacteria whose LAD score exceeded the default value of 2. Red represents the ABOi-4 group, and blue represents the ABOc-4 group. (**H**) Top 6 significantly different Level 3 KEGG pathways between the two groups. (**I**) PCA map, where the distance of each coordinate point highlights the degree of aggregation and dispersion between the samples. (**J**) OPLS-DA map, where the abscissa is the predicted principal component representing the difference between the groups, and the ordinate was the orthogonal principal component, representing the difference within the group. (**K**) Volcano plot of the differential metabolites between the two groups. (**L**) KEGG pathway bubble chart of bile acid, where the abscissa is the Rich factor, and the larger the value, the greater the enrichment. The ordinate signifies the metabolic pathway with the highest degree of enrichment. The color of the dots stands for the *p*-value, and the redder it is, the smaller the *p*-value, indicating that the enrichment was more obvious. The size of the dot represents the number of differential genes in the pathway, and the larger the dot, the greater the number of differential genes. * *p* < 0.05, *** *p* < 0.001.

**Figure 5 jcm-12-00141-f005:**
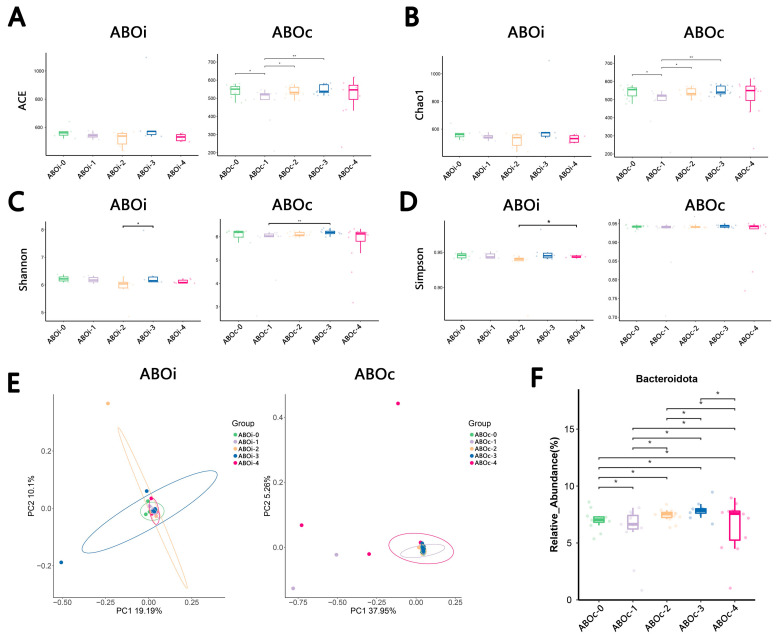
Longitudinal analysis of the biliary microbiota in ABOi LT and ABOc LT. Contrast of the alpha diversity variation with the use of (**A**) ACE, (**B**) Chao1, (**C**) Shannon and (**D**) Simpson. (**E**) The results of the PCoA variation of the biliary microbiota in ABOi LT and ABOc LT. (**F**) Changes in the relative abundance of *Bacteroidetes* in ABOc LT. * *p* < 0.05, ** *p* < 0.01.

**Table 1 jcm-12-00141-t001:** Clinical patient characteristics.

	ABOi LT Group	ABOc LT Group	*p*-Value
Patients, n	6	12	NA
Male, n (%)	6 (100%)	11 (91.7%)	1.000
Age (years)	48 (32–59)	46 (24–56)	0.424
BMI (kg/m^2^)	21.75 (18.9–26.3)	22.31 (16.9–26.1)	0.574
Previous abdominal surgery, n (%)	1 (16.7%)	2 (16.7%)	1.000
WBC (10^9^/L)	11.5 (7.1–11.3)	4.8 (0.9–12.5)	0.007
Hb (g/L)	95 (84–106)	86 (62–111)	0.146
Blood Plt (10^9^/L)	128 (51–212)	46 (19–97)	0.003
CRP (g/dL)	27.2 (21.7–51.2)	8.7 (<5–20.7)	0.004
ALB (U/L)	33.9 (28.3–37.9)	33.8 (30–42.2)	0.639
ALT (U/L)	73 (8–200)	38 (11–130)	0.190
AST (U/L)	150 (30–534)	54 (18–90)	0.083
GGT (U/L)	137 (18–438)	40 (15–107)	0.223
ALP (U/L)	220 (45–798)	107 (44–186)	0.606
TBIL (μmol/L)	263 (37–594)	276 (21–638)	0.851
Cr (μmol/L)	112 (39–346)	80 (56–138)	0.925
MELD score	28 (18–36)	26 (10–37)	0.673
Warm ischemia time (min)	3.3 (3–4)	3.3 (3–5)	0.811
Cold ischemia time (min)	62.8 (42–79)	42.3 (29–58)	0.006
Operation time (min)	343 (229–420)	287 (207–366)	0.061
Post-transplantation ICU stay time (days)	16 (10–24)	14 (5–36)	0.477
Donor age (years)	50 (36–64)	45 (11–59)	0.605
Donor male, n (%)	5 (83.3%)	11 (91.7%)	1.000
Donor BMI (kg/m^2^)	24.72 (20.8–30.4)	22.83 (14.7–27.4)	0.454
GRWR (%)	1.59 (1.18–2.05)	1.70 (1.05–2.40)	0.708

BMI: body mass index; WBC: white blood cell; Hb: hemoglobin; Plt: platelet; CRP: C-reactive protein; ALB: albumin; ALT: alanine aminotransferase; AST: aspartate aminotransferase; GGT: gammaglutamyl transpeptidase; ALP: alkaline phosphatase; TBIL: total bilirubin; Cr: creatinine; MELD: model for end-stage liver disease; ICU: intensive care unit; NA: not available; GRWR: graft-to-recipient body weight ratio.

## Data Availability

The data presented in this study are available upon request from the corresponding author.

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
