# Peer review of "ABO-Incompatible Liver Transplantation under the Desensitization Protocol with Rituximab: Effect on Biliary Microbiota and Metabolites"

_jcm, 2022, doi:10.3390/jcm12010141_

Round 1

Reviewer 1 Report

The present manuscript by Xiao et al. analyzed the bile microbiota of 6 patients with ABO incompatible LT and 12 patients with ABO compatible LT at several timepoints within the first month after surgery. The authors concluded that the bile microflora is stable during this period irrespective of the donor/recipient ABO and the use of pre-LT conditioning therapy. The topic could be of interest but the sample size is clearly insufficient to provide reliable conclusions. In addition, the authors did not evaluate the relationship between bile microbiota and any clinically relevant outcome (rejection, graft failure, biliary complications…).

The authors are kindly invited to consider the following comments:

-      ***    The main limitation of the study is the lack of clinical applicability. The main aim of the study is to analyze alterations of the bile microflora but it is unclear whether an altered bile microflora would translate into worse outcomes in liver transplant patients. The authors did not follow the patients to assess clinically relevant outcomes such as rejection, biliary complications, graft loss…

-     ***     Sample size calculation is missing. Taking into account the expected heterogeneity in the bile microflora among individuals, sample size may be insufficient. There are only 6 patients in the ABOi group. The authors are encouraged to increase the sample size after realistic estimations and to prolong surveillance to assess clinical outcomes.

-     ***     The are too many comparison groups, all of them with n<15. Results are underpowered and not significant comparisons, which are the vast majority in the study, are not reliable.

-    ***      Baseline samples were taken during surgery through gallbladder aspiration whereas post-LT samples were taken from aspiration through the T tube. This different protocol could influence isolated bacteria. The presence of the T tube itself could influence bacterial populations, particularly increasing gram-positives.  

-    ***      The abstract should clearly state the number of patients (not only samples) in each comparison arm.

-      ***    There is a typo in the results section (line 242): ABOi for ABOc.

Author Response

  1. The main limitation of the study is the lack of clinical applicability. The main aim of the study is to analyze alterations of the bile microflora but it is unclear whether an altered bile microflora would translate into worse outcomes in liver transplant patients. The authors did not follow the patients to assess clinically relevant outcomes such as rejection, biliary complications, graft loss…

Author Response: We have addressed the above point. This was a pilot study to explore the effects of ABOi LT on biliary microbiota. Among the 18 LT recipients, only one patient in the ABOi LT group developed biliary fistula at 1 month, and the rest had no adverse clinical events within 1 month. Please see line 249 of the revised manuscript.

  1. Sample size calculation is missing. Taking into account the expected heterogeneity in the bile microflora among individuals, sample size may be insufficient. There are only 6 patients in the ABOi group. The authors are encouraged to increase the sample size after realistic estimations and to prolong surveillance to assess clinical outcomes.

Author Response: As ABOi LT is performed in an emergency setting, ABOc LT is preferred for patients. Therefore, the sample size of ABOi LT was small.

  1. The are too many comparison groups, all of them with n<15. Results are underpowered and not significant comparisons, which are the vast majority in the study, are not reliable.

Author Response: The purpose of this study is to compare the biliary microbiota of the two transplantation methods from different cross sections and to dynamically observe the changes of biliary microebiota over time after transplantation. The number of patients in each group was too small due to the low sample size.

  1. Baseline samples were taken during surgery through gallbladder aspiration whereas post-LT samples were taken from aspiration through the T tube. This different protocol could influence isolated bacteria. The presence of the T tube itself could influence bacterial populations, particularly increasing gram-positives.

Author Response: We could not obtain a preoperative bile sample because the bile had to be obtained by an invasive procedure, and T-tube is also the only method currently available to study the biliary microbiota of liver transplantation, according to previous studies, so it is the best way method we known even if it affects the results. In order to obtain samples of biliary microbiota, it is necessary to further explore the methods

  1. The abstract should clearly state the number of patients (not only samples) in each comparison arm.

Author Response: We have added the above to the abstract. Please see line 46 of the revised manuscript.

  1. There is a typo in the results section (line 242): ABOi for ABOc.

Author Response: We have addressed the above point in the revised manuscript.

Reviewer 2 Report

In this study, Xiao et al. seek the effects of liver transplantation with ABO-compatible or incompatible liver grafts with rituximab treatments. This study is poorly designed and missing critical factors.

Previous studies showed that ABO blood types can influence microbiota. It is unclear if blood type can alter biliary microbiota, and if it is different when patients with different blood types have incompatible transplantation. Blood types not only in recipients, but also in donors might change gut, intestinal, or biliary microbiota. In Table 1, there are significant differences between ABOi and ABOc groups in WBC, Blood Plt, CRP, and Cold ischemia time. All these factors, numbers of white blood cells and platelets, C-reactive protein, and ischemia and reperfusion injury can alter microbiome, shown in previous studies. The authors ignore these factors completely in this study. It is impossible to conclude that the differences the authors identify in this study are due to incompatible transplantation, donor/recipient blood type, or numbers of platelets. Same problem for rituximab. Rituximab can alter microbiome, but the authors do not compare data between patients with or without rituximab, so the effects of rituximab are unclear. This study does not show any finding or prove anything.

Author Response

  1. In Table 1, there are significant differences between ABOi and ABOc groups in WBC, Blood Plt, CRP, and Cold ischemia time. All these factors, numbers of white blood cells and platelets, C-reactive protein, and ischemia and reperfusion injury can alter microbiome, shown in previous studies. The authors ignore these factors completely in this study.

Author Response: Because patients receiving ABOi LT are in the acute phase of the disease and have worse liver function, it is inevitable that there are differences in white blood cells, platelets, and CRP between ABOi LT and ABOc LT. There are many factors that affect the microbiota, and it is difficult to achieve a unified baseline of all factors.

  1. Rituximab can alter microbiome, but the authors do not compare data between patients with or without rituximab, so the effects of rituximab are unclear.

Author Response: The effect of rituximab on microbiota has only been verified in mice, and its effect in human microbiota is still unclear, which needs to be verified by further studies. Since the current ABOi LT cannot be performed without the use of rituximab, we can consider ABOi LT and rituximab as common variables in the ABOi LT group.

Reviewer 3 Report

This paper is well-done totally. I just asked a complete revision by a native speaker and microbiologist. Many errors to describe the genus or phylum name. The word, such as microflora, was seldom used currently.

Author Response

This paper is well-done totally. I just asked a complete revision by a native speaker and microbiologist. Many errors to describe the genus or phylum name. The word, such as microflora, was seldom used currently.

Author Response: We have addressed the above point. Please see line 58 and line 467 of the revised manuscript.

Reviewer 4 Report

This study was so valuable in terms of future research. I have some question.

1.       Their sample size was a relatively small, so it was difficult to draw their conclusion.

2.       What was the reason why there were no differences regarding the biliary microbiota and metabolites between two group?

3.       What was the reason why there was a difference at only postoperative two days after transplantation?

4.       Was the information of surgical information (transfusion information) and preoperative original disease such as HCC unnecessary?

Author Response

  1. Their sample size was a relatively small, so it was difficult to draw their conclusion.

Author Response: Due to the current situation of LT, the sample size is small.

  1. What was the reason why there were no differences regarding the biliary microbiota and metabolites between two group?

Author Response: There are two possible reasons: First, ABOi LT itself does not cause changes in the biliary microbiota; Second, ABOi LT causes changes in the biliary microbiota, according to the prognosis of ABOi LT is improved under the desensitization effect of rituximab, and it may be the effect of rituximab that does not cause significant changes in biliary microbiota. All these need to be verified by follow-up studies. Please see line 588 of the revised manuscript.

  1. What was the reason why there was a difference at only postoperative two days after transplantation?

Author Response: β-diversity analysis indicated that binary–jaccard dissimilarity is different in ABOi LT and ABOc LT (P = 0.046). But the other three indices showed no difference. We believe that the wrong p-value of the binary–Jaccard may be caused by the small sample size. And α-diversity analysis indicated increased average biodiversity of the bile fluid microbiota in group ABOi-1. The possible reason is that ABOi LT recipients are in a more acute course of the disease and have a more severe systemic inflammatory response than ABOc LT recipients. Please see line 512 and 520 of the revised manuscript.

Round 2

Reviewer 1 Report

The authors implemented minor changes but the two main limitations of the study remain: reduced sample size (without providing a rationale for sample size calculation) and absence of clinical applicability of the findings of the study. I acknowledge that these are inherent limitations which may not be overcome with further manuscript changes. The whole study would need to be re-designed and powered to aim at clinically relevant outcomes.

Author Response

The authors implemented minor changes but the two main limitations of the study remain: reduced sample size (without providing a rationale for sample size calculation) and absence of clinical applicability of the findings of the study. I acknowledge that these are inherent limitations which may not be overcome with further manuscript changes. The whole study would need to be re-designed and powered to aim at clinically relevant outcomes.

Author Response: Thank you very much for your valuable suggestions. We will carry out further work on these issues in the future.